# The Effect of Oxidative Stress on the Human Voice

**DOI:** 10.3390/ijms25052604

**Published:** 2024-02-23

**Authors:** Shigeru Hirano, Haruhiko Inufusa, Fukka You

**Affiliations:** 1Department of Otolaryngology Head and Neck Surgery, Kyoto Prefectural University of Medicine, 465 Kajii-cho, Kamigyo-ku, Kyoto 602-8566, Japan; 2Division of Antioxidant Research, Gifu University, Gifu 501-1194, Japan; hinufusa@gmail.com (H.I.); y@antioxidantres.jp (F.Y.)

**Keywords:** oxidative stress, voice, vocal fold, antioxidant

## Abstract

The vocal fold vibrates in high frequency to create voice sound. The vocal fold has a sophisticated histological “layered structure” that enables such vibration. As the vibration causes fricative damage to the mucosa, excessive voicing can cause inflammation or injury to the mucosa. Chronic inflammation or repeated injury to the vocal fold occasionally induces scar formation in the mucosa, which can result in severe dysphonia, which is difficult to treat. Oxidative stress has been proven to be an important factor in aggravating the injury, which can lead to scarring. It is important to avoid excessive oxidative stress during the wound healing period. Excessive accumulation of reactive oxygen species (ROS) has been found in the injured vocal folds of rats during the early phase of wound healing. Antioxidants proved to be useful in preventing the accumulation of ROS during the period with less scar formation in the long-term results. Oxidative stress is also revealed to contribute to aging of the vocal fold, in which the mucosa becomes thin and stiff with a reduction in vibratory capacity. The aged voice can be characterized as weak and breathy. It has been confirmed that ROS gradually increases in rat vocal fold mucosa with age, which may cause further damage to the vocal fold. Antioxidants have also proved effective in avoiding aging of the vocal fold in rat models. Recently, human trials have shown significant effects of the antioxidant Twendee X for maintaining the voice of professional opera singers. In conclusion, it is suggested that oxidative stress has a great impact on the damage or deterioration of the vocal folds, and the use of antioxidants is effective for preventing damage of the vocal fold and maintaining the voice.

## 1. Introduction

The voice is essential in human life not only as a communication tool but also for vocal arts such as singing. The voice is produced by the vibration of the vocal folds located in the voice box (larynx) [1,2,3]. The vocal folds are a pair of mucosae about 1.5 to 2 cm in length [3]. They abduct during respiration to open the glottis and adduct during production of voice to close the glottis. The vocal folds vibrate via the air flow during exhalation. The vocal fold consists of mucosa and vocalis muscle. The mucosa is only 1 mm thick but has three layers: the superficial layer of the lamina propria (SLP) and intermediate and deep layers [1,2,3]. The intermediate layer is composed of elastic fiber, and the deep layer is supported by collagen fiber. The intermediate and deep layers form the vocal ligament. The vocal ligament has not been observed in any mammal except for humans [4]. It does not exist even in human childhood. It is developed with age and finally matures during puberty when voice mutation is completed [5,6,7]. The role of the vocal ligament is still unclear, but it is suggested that the ligament is beneficial during high-pitch phonation to support the high tension of the mucosa like a high-pitch string of a guitar. This may be one of the reasons for the wide pitch range of the human voice. The vocalis muscle provides a firm scaffold by muscular contraction. The SLP has rare fibrous proteins but contains a range of amorphous substances including glycosaminoglycan (GAG), proteoglycan, and adhesive molecules [8,9]. Hyaluronic acid (HA) is one of the GAGs and has been considered as the most important molecule that is essential in maintaining ideal vibration of the mucosa. Indeed, it was revealed that the vocal fold did not vibrate after HA was completely removed from the vocal fold mucosa [8]. Thus, the SLP is pliable and the center part of vibration. The SLP vibrates in super-rapid motion as fast as 100 cycles per second in men, 200 cycles in women, and up to 800 cycles in soprano singers. This vibratory function is typical for the vocal fold mucosa, and there is no other mucosa that can vibrate like it. The vibration of the SLP is supported by the contraction of the vocalis muscle. This structure is called “cover body theory” [2], in which the vocalis muscle serves as the body and the SLP vibrates as the cover.

As described above, the vocal fold mucosa is so thin and delicate that it can be damaged easily by overuse (vocal abuse), inflammation caused by upper respiratory infection, smoking, inhalation of chemical pollution, or allergy as well as direct injury by burn, radiation, or surgery to the vocal fold [10,11]. The damage can cause emission of reactive oxygen species (ROS), which induces oxidative stress, and can be aggravated by the ROS. ROS also increases throughout the whole body with the aging process, and can cause cancer, diabetes, vascular diseases, pulmonary disease, and other illnesses [12]. The vocal fold is not the exception, and it is usually altered with age histologically and functionally. It is essential to control ROS to maintain the vocal fold and voice.

To date, there have been increasing data about the role of oxidative stress on the human voice, and we have reviewed the literature related to the topic in this article in terms of how ROS affects the vocal fold and how antioxidant therapy works to maintain the human voice.

## 2. Role of Oxidative Stress on Vocal Pathologies

Vocal fold pathologies include polyp, nodule, cyst, and Reinke’s edema (RE) [13,14,15,16]. A vocal fold polyp is usually a unilateral lesion that occurs as a result of focal hemorrhage inside the mucosa. Loud phonation or strong coughing can cause the formation of a polyp. Vocal fold nodules are bilateral lesions that occur at the anterior one third of the vocal fold, called the “striking zone” because it is exposed to the maximum contact pressure during vocal fold vibration. Chronic injury creates a small bilateral bump. Women and children are susceptible to developing vocal fold nodules because the pitch of vibration of their voices is high, which results in frequent injury of the striking zone. Vocal fold cysts are created inside the SLP after injury to the mucosa, possibly because the epithelium at the injured site moves inside the SLP during wound healing, creating the cyst wall. RE is featured with edema in the SLP, which is usually caused by chronic inflammation due to smoking. These pathologies are caused by inflammation or injury to the vocal fold mucosa, and it is suspected that ROS may have an important role in the pathogenesis of these pathologies. Indeed, there have been several studies including clinical studies that suggest the pathogenic contribution of oxidative stress on vocal fold pathologies.

Branski et al. [17] reported excessive accumulation of ROS in Reinke’s edema in human patients. They examined the effects of cigarette smoke exposure to the vocal fold tissue using viable porcine vocal fold and human vocal fold fibroblasts. Cigarette smoke condensate was applied to both tissues and cells, and the results indicated that transepithelial resistance was preserved; however, the gene expression of cyclooxygenase 2 (COX-2) and its downstream lipid mediator prostaglandin E2 (PGE2) were upregulated by cigarette smoke exposure. It was suggested that cigarette smoke initiates an inflammatory response in vocal fold fibroblasts, but the vocal fold mucosa may have a durable epithelial barrier function.

Alper et al. [18] also exposed freshly excised, viable porcine vocal fold epithelium to hydrogen peroxide for 2 h and reported that exposure to ROS did not significantly alter transepithelial resistance, although a small trend for a decreased concentration of epithelial junctional complex protein was observed. These findings indicate that oxidative stress can alter the vocal fold epithelial function, while the vocal fold has some antioxidant barrier function. RE may be the consequence of the balance and interaction between ROS and antioxidative function in the vocal fold.

Gugatschka et al. [19] performed proteomic analyses of human vocal fold fibroblasts cultured in a medium conditioned with cigarette smoke extract to reveal the mechanism of Reinke’s edema. The proteomic analyses revealed that cigarette smoke increased the quantity of proteins involved in oxidative stress responses, and genes linked to ROS were enriched in the cigarette-smoke-induced proteins. Furthermore, they found downregulation of genes of fibrillar collagen, COL1A1 and COL1A2, and upregulation of UDP-glucose 6-dehydrogenases, an inhibitor for the biosynthesis of hyaluronic acid, which means increased deposition of HA in RE. It was suggested that RE may be created by an increase in HA and decrease in collagen fibrils via oxidative stress responses.

“Laryngitis” is the term widely used for inflammation of the larynx, and it is caused by viral or bacterial infection, smoking, laryngopharyngeal reflux, and air pollution. Particulate matter (PM) is a major component of air pollution that can affect the airways, including the vocal fold. Recently, the relationship between laryngitis and PM was reported in a few epidemiological studies [20], but the pathophysiology was unclear. Choi and Kim [21] examined the effects of PM exposure on human vocal fold fibroblasts using an in vitro study. They showed that ROS was significantly increased by the exposure to PM using a dichloro-dihydro-fluorescein diacetate (DCFH-DA) fluorescent dye probe with significant increases in 4-HNE, a marker for lipid peroxidation, and 8-OHdG, a marker for oxidative DNA damage, in immunohistochemical examinations. They also found a significant increase in AhR, an important inflammatory modulator, and CYP1A1 with significant increases in IL-6 and IL-8, the pro-inflammatory cytokines. These findings indicate that PM induces ROS formation and pro-inflammatory cytokines via the AhR-CYP1A1 pathway and causes lipid peroxidation and DNA damage. It is suggested that the ROS-related cell damage and inflammatory response in vocal fold fibroblasts may be a mechanism underlying chronic laryngitis.

## 3. Vocal Fold Wound Healing and Scarring

Given that the vocal fold mucosa is tiny and delicate with a fine histological structure, vocal fold injury or inflammation causes severe dysphonia. The sequel of injury or chronic inflammation is the development of a polyp, nodule, cyst, or edematous lesion, and the worst case is the scarring of the vocal fold. Vocal fold scarring is featured with stiffened mucosa, which leads to a reduction in or the absence of vocal fold mucosal vibration, resulting in permanent severe dysphonia [10,11]. In general, scarring is regarded as fibrous tissue with an increase in collage type I, which strengthens the rigidity of the tissue. Vocal fold scars contain dense collagen deposition in the SLP with other ECM alterations. Hirano et al. [22] examined the histology of the vocal folds after cordectomy for early glottic cancers and found mild to severe scarring in the SLP, which is featured with excessive collagen deposition and a reduction in HA. The vibration was worse in the case of severe scarring. Once scarred, it is difficult to treat the scarred vocal folds; thus, the prevention of scar formation is very important.

Vocal fold sulcus or “sulcus vocalis” is another pathology with stiffened vocal fold mucosa [23]. A sulcus is created at the mid-membranous portion of the vocal fold mucosa along the antero-posterior axis. Vocal fold sulcus is classified into three types [24]. Type I sulcus is a naturally occurring sulcus that causes little dysphonia, called “physiological sulcus”. Type II sulcus is a deeper sulcus in which the bottom of the sulcus loses the SLP and the epithelium directly attaches to the vocal ligament. Type III sulcus has the deepest sulcus, which penetrates the vocal ligament. The etiology of sulcus is controversial: congenital, acquired, or both. Nakayama et al. [25] suggested that the inflammatory process may cause sulcus formation. Histological examination showed a thick collagen bundle at the bottom of the sulcus, which is similar to vocal fold scarring [26].

The wound healing mechanism includes three phases [27,28]: the inflammatory phase, proliferative phase, and remodeling phase. The inflammatory phase occurs immediately after injury and continues for about 3 days. The key functions of inflammatory phase are hemostasis and the initiation of inflammation [28]. Platelets accumulate at the injury site and trigger the inflammatory response by releasing growth factors, such as PDGF and TGF, and cytokines [29]. Neutrophils first appear within hours and macrophages predominate by 48–72 h. They act to phagocyte necrotic debris and bacteria [30]. Several animal studies have been reported to explore the mechanism of vocal fold wound healing [31,32,33,34,35,36,37,38]. Branski et al. [31] developed a rabbit model of vocal fold injury and reported that a fibrous clot appeared on day 1, massive cellular infiltration occurred on day 3, and re-epithelialization was completed on day 5. The proliferation of fibroblasts was observed on day 3. In general, macrophages appear on day 1 through day 7 [28,29]. Pro-inflammatory macrophages (M1) appear first and are then replaced with anti-inflammatory macrophages (M2) over time [34]. HA has been reported to increase immediately after wounding of skin, but HA was decreased during the early wound healing period of the vocal fold in rabbit and pig models [39]. Ohno et al. [38] examined the gene expression of hepatocyte growth factor (HGF) and transforming growth factor (TGF)-b1 during the early phase of rat vocal fold wound healing and found an immediate increase in TGFb1 followed by a later increase in HGF. HGF is a strong anti-fibrotic factor and TGFb1 is a strong pro-fibrotic factor [40]. The balance of these factors seems to be important for long-term remodeling of ECM in the SLP.

The proliferative phase follows or overlaps the inflammatory phase at around day 3 through 1 month [41]. It is featured alongside mesenchymal cell proliferation, angiogenesis, and epithelialization. Fibroblasts infiltrate at 48–72 h after wounding, leading to granulation formation [42]. The proliferation is stimulated by PDGF, TGFb, and basic FGF [42]. Fibroblasts begin to produce many ECM components including collage type I, III, elastin, fibronectin, and GAGs to repair the wounded tissues. Angiogenesis occurs within days, being stimulated by bFGF, VEGF, and TGF [41]. TGFb1 stimulates the transition of fibroblasts to myofibroblasts [43]. Myofibroblasts contribute to scar formation by producing massive collagen. The remodeling phase initiates around 1 month following the proliferative phase. During the period, ECM turnover takes place and continues for up to 1 year. ECM turnover, especially collagen, is primarily mediated by the balance between MMPs (metamorphoproteinase) and TIMPs (tissue inhibitor of MMPs) [44]. Maturation of scarring is featured with completion of collagen deposition, and it has been reported that vocal fold scarring is matured at 2 months in rats [45] and 6 months in rabbits and dogs [46,47]. Kishimoto et al. [48] followed up patients after cordectomy for glottic cancers and found stabilization of phonatory function at around 6 months after the surgery.

## 4. The Role of Oxidative Stress and Antioxidants on Vocal Fold Wound Healing

As mentioned above, the wound healing mechanism is very complicated, being affected by a myriad of factors including several types of cells, cytokines, growth factors, and ECMs (Figure 1). ROS is critically important during wound healing because it can affect every factor. In general, ROS has an important role in the protection of the wound site from bacteria during the wound healing period [49]. However, it was reported that over-exposure to ROS impairs wound closure [50,51].

Mizuta et al. [52] established a rat vocal fold injury model by incising the mucosa transorally. Histology showed infiltration of inflammatory cells during day 1–3, and a firm scar was finally created on day 56. ROS rapidly increased during day 1–3 and quickly disappeared after day 5. The inflammatory cells including macrophages and neutrophils were found to produce ROS during the early phase. It was suggested that ROS during early wound healing may lead to long-term scarring of the vocal fold tissue. Then, they examined the effects of antioxidant therapy on vocal fold wound healing using the same rat model [53]. They applied transoral astaxanthin (AST) from one day before the injury through seven days after the injury. AST is a carotenoid that has proved to have a strong antioxidant effect. The results showed that ROS on day 1 and 3 was significantly suppressed, and scar formation on day 58 was less extensive than that without AST. It was suggested that antioxidants during the early phase of wound healing could prevent scarring of the vocal folds.

AST is a xanthophyll carotenoid [54] that has been proven to modulate oxidative stress and inflammation through a reduction in free radicals and activate endogenous antioxidant systems via genetic modulation [55,56]. More recently, positive effects of AST have been shown on wound healing of the nasal mucosa and impaired skin regeneration [57,58,59]. Manciula et al. [57] damaged the nasal mucosa in rats using the brushing method and treated them using astaxanthin or dexamethasone. The epithelial thickness index (ETI) and the subepithelial thickness index (STI) were significantly lower in the AST-treated group, and the goblet cell count was higher in the AST group. They concluded that AST significantly decreased fibrosis, inhibited synechia development and significantly decreased subepithelial fibrosis with no general or local toxic effects. Meephansan et al. [58] examined the effects of topical AST treatment for full-thickness dermal wounds in mice. AST-treated wounds showed noticeable contraction by day 3 of treatment and complete wound closure was observed by day 9, while the wounds without AST application showed only partial epithelialization and still carried scabs. Col1A1 and bFGF were significantly increased in the AST group. The iNOS, an oxidative stress marker, showed a significantly lower expression in the AST-treated group. The results indicated that AST is an effective material for accelerating wound healing.

## 5. Aging of Vocal Fold and Voice

The voice changes with age. It has been reported that up to 30% of people over 65 years old have a voice problem that compromises their QOL [60,61]. The aged voice is featured with altered acoustic properties, increased roughness, and increased vocal instabilities [62].

One of the main reasons for the aging voice is the histological change in the vocal fold. Histological studies using cadaveric human larynges have demonstrated the increase in collagen fibers forming thick bundles, the decrease in or disorganization of elastic fibers, and decrease in HA in the lamina propria of the mucosa [62,63,64,65,66,67,68]. As a result, the SLP becomes thinner and stiffer. Animal studies using rats clearly showed a gradual increase in collagen deposition and decrease in HA with age in the vocal fold mucosa [69]. It was also revealed that the vocalis muscle is altered with age, which resulted in atrophy of the muscle [70].

Fibroblasts are the key cells that maintain the vocal fold mucosa by producing every kind of ECM, MMP, and TIMP, which are essential for turnover of ECMs and for maintaining the healthy tissues [71,72]. It is confirmed that healthy fibroblasts contain rough endoplasmic reticulum (rER) and golgi apparatus (GA) in the cytoplasm, which means active function of protein production. Vitamin-A stored fibroblasts have also been found in the vocal fold. Hirano et al. [72] reported that geriatric human vocal fold fibroblasts showed less rER or GA with glycogen particles and lipofusion granules, which indicated the deterioration of the cell and cell function. Ding et al. [73,74] examined senescent expression of genes coding collagens, collagen-degrading metalloproteinases, and tissue inhibitors of metalloproteinases as well as tropoelastin, elastase, and lysyl oxidase using old rat vocal folds. They found a decrease in mRNA expression of collagen I, V, and collagenase, which was interpreted as a slowdown of collagen turnover. Gene expression of tropoelastin showed no age-related change, but the expression of elastase was increased. Ohno et al. [69] also performed a gene expression study using old rats and revealed down-regulated expression of procollagen types I and III, matrix metalloproteinases 2 and 9, and HA synthases 1, 2, and 3 in elderly vocal folds compared to young vocal folds. 

These genetic and histologic age-related changes in the vocal fold cause age-related dysphonia [75,76,77,78,79]. The atrophy of the vocal fold mucosa leads to bowing, glottic incompetence, reduced periodicity, and reduced mucosal wave amplitude. Many investigators have reported a decrease in aerodynamic function, alteration of acoustic signals, increase in voice handicap index, and increase in perceptive score of the voice [75,76,77,78,79,80]. Laryngeal electromyography (EMG) [81,82] indicated reduced amplitude and decreased firing rates in the vocalis muscle in human subjects. A decrease in muscular function causes a decrease in loudness and stability of voice.

## 6. Role of Oxidative Stress and Antioxidant on Aging Voice

The role of oxidative stress has been researched in the aging process of the vocal fold. As mentioned above, it was confirmed that aged rat vocal folds showed a decrease in HA and over-deposition of collagen fibers in the SLP. Mizuta et al. [83] confirmed a significant increase in ROS in the SLP in aged rats, suggesting that accumulation of ROS may cause the aged alteration of the SLP. They fed rats with or without astaxanthin (AST) from 6 months of age to 18 months. A total of 70%of the rats survived until the age of 18 months in the AST group, while only 50% of the rats survived in the control group without AST. ROS marked by 4-HNE was significantly increased in the control aged group as compared to young rats, but the increase in ROS in the AST group was significantly lowered. HA significantly reduced in the aged group, but no significant decrease was observed in the AST group. Gene expression analyses indicated no significant differences in the mRNA expression of bFGF and HGF between the AST group and the young group, although the expression of those genes was significantly reduced in the aged control group as compared to the young group. One possible mechanism of AST for anti-aging of the vocal fold may be due to the maintenance of bFGF and HGF, because both growth factors have been proven to stimulate HA production from the vocal fold fibroblasts [84]. They concluded that ROS is one of the most important factors to proceed the aging process of the vocal fold, and antioxidants can slow the aging process.

## 7. Antioxidant for Maintenance of Voice

Vocal abuse, which means too much voicing, causes inflammation and injury to the vocal folds, and chronic vocal abuse can result in scarring [85,86,87]. Swanson et al. developed an in vivo vocal abuse model of rabbits [85]. The animals received 30 min of experimentally induced modal or raised-intensity phonation, and gene expression of inflammatory cytokines was determined using real-time PCR. The results showed a significant increase in gene expression of IL-1beta, TGFbeta1, and COX-2 during raised-intensity phonation compared to modal phonation. Kojima et al. also reported a significant reduction in the microprojection density, microprojection height, and depth of the epithelial surface with increasing time and phonation magnitudes using the same model, which suggested the increased epithelial surface damage risk with excessive and prolonged vibration exposure [86]. Rousseau et al. further examined the gene expression of tight junction proteins and revealed significantly decreased occludin and β-catenin gene expression in the rabbits that underwent raised-intensity phonation compared with control. Scanning electromicroscopy revealed significant obliteration, desquamation, and evidence of microhole formation in rabbit vocal folds exposed to raised-intensity phonation compared with the control. These results have suggested that a transient episode of raised intensity phonation alters the transcript levels of vocal fold intercellular tight junction proteins and disrupts the epithelial barrier [87].

The damage to the vocal fold can occur particularly in professional voice users including singers, actors, entertainers, etc. As such inflammation or injury causes excessive production of ROS, antioxidants are expected to help avoid the damage to the vocal fold and maintain the function. Kaneko et al. [88] conducted a clinical trial to examine the protective effects of antioxidants during a vocal loading task. Ten normal male subjects with an age ranging from 23 to 30 years old were involved in the study. They underwent a 60-min vocal loading session and received vocal assessments prior to, immediately after, and 30 min post vocal loading. All subjects were then supplied oral astaxanthin for 28 days, and then they received the same vocal task and assessments. Phonatory function including aerodynamic, acoustic, and perceptive scores at the first session without AST showed significant worsening immediately after vocal loading, but recovered in 30 min after loading. When AST was applied in advance, none of the phonatory parameters became significantly worse, even immediately after vocal loading. The results suggest that antioxidants may be useful for protecting the voice during vocal abuse.

Hirano et al. [89] further examined the effects of the antioxidant Twendee X on maintenance of the singing voice in professional opera singers. Twenty-six opera singers were involved in this study. Their ages varied from 34 to 77 years old (mean 51 years), and 12 singers were male while 14 singers were female. Twendee X (TWX) is a multi-ingredient supplement including vitamin C and seven other elements, and it has been proven to have strong antioxidant effects. TWX was applied to the singers for 1 month and the singing voice handicap index-10 (SVHI-10) was assessed before and after the application of TWX. The SVHI-10 was significantly decreased in all singers, indicating better performance of the singing voice. The highest improvement was observed in vocal fatigue during singing and the capability to maintain singing. The results indicated the effects of TWX for maintenance of the singing voice in the professional signers.

TWX is a strong antioxidative supplement consisting of eight ingredients: L-glutamine, ascorbic acid, L-cystine, coenzyme Q10, succinic acid, fumaric acid, riboflavin, and niacin amid. It has been proven that TWX has higher antioxidant power than vitamin C by about five-fold. You et al. [90] conducted an in vitro study in which human hepatocellular carcinoma cells, HepG2 cells, were treated with H_2_O_2_. H_2_O_2_ induced ROS in the cells and mitochondria, and TWX significantly reduced the H_2_O_2_-induced ROS. TWX also increased the activity of SOD, an antioxidant enzyme, and reduced the ratio of oxidized glutathione (GSSG) to reduced glutathione (GSH), indicating its ability to protect cells and mitochondria from oxidation. Lately, a multicenter, randomized, double-blind, placebo-controlled, prospective intervention trial indicated the significant effects of TWX to prevent progression of dementia in patients with mild cognitive impairment (MCI) [91]. TWX is expected to maintain the function of several organs and tissues including the vocal fold, although the specific mechanism of how TWX works for the vocal fold is not known, and future research is warranted to resolve this aspect.

## 8. Conclusions

Maintenance of the vocal fold and voice is important for maintaining QOL in daily life and for healthy aging. Although an effective strategy has not been fully developed, oxidative stress should be the main target. Various evidence confirms the roles of ROS in the deterioration and aging of the vocal fold. Antioxidants may be the key to preventing long-term damage and scarring of the vocal fold in cases of vocal abusers such as singers. Antioxidants are also regarded to contribute to anti-aging of the vocal folds.

There have been several animal studies that support the effects of antioxidants for the prevention of vocal fold damage or aging and for maintaining the voice. However, the clinical evidence has not been fully accumulated. Randomized control clinical studies are warranted to find evidence of the effectiveness of antioxidants for maintenance of the voice. It is also needed to clarify how oxidative stress occurs in the vocal folds according to the degree of vocal abuse, damage, or inflammation. The establishment of animal models is required to explore this aspect. Furthermore, the contribution of oxidative stress and antioxidant on cancerization should be examined to develop steady-state preventative strategies of laryngopharyngeal cancers, which will become more important in aging societies.

## Figures and Tables

**Figure 1 ijms-25-02604-f001:**
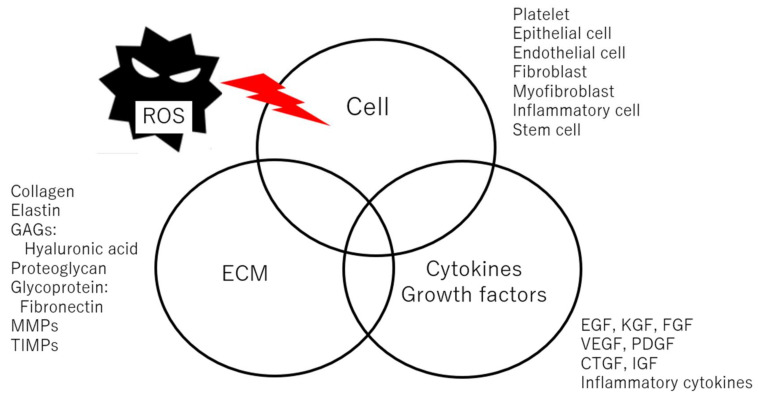
Factors during wound healing.

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
