# Peer review of "The Effect of Oxidative Stress on the Human Voice"

_ijms, 2024, doi:10.3390/ijms25052604_

Round 1

Reviewer 1 Report

Comments and Suggestions for Authors

I am very pleased to review the article however some of following should included next revision

1. change the tittle, should role of oxidative stress and voice in human, author can change such impact of oxidative stress and voice on humans

2. ABstract should be small 

line 49-50 there is no meaning of sentense, change or clarifiy

4. add few line in introduction about explaining review article details

5. section 5 and 6 add few related article how role of antioxidant in human body, author should explain mechanism of oxidative stress in human voice with citiation and role of antioxidant agent on human body particularly vocal cord

Author Response

Thank you very much for several thoughtful comments and suggestions for our paper, all of which are really helpful and instructive.

  1. change the tittle, should role of oxidative stress and voice in human, author can change such impact of oxidative stress and voice on humans

[Response] Thank you for the suggestions and we changed the title to “Role of oxidative stress on voice in human”

  1. ABstract should be small 

[Response] We shortened the abstract by deleting unnecessary parts.

  1. line 49-50 there is no meaning of sentense, change or clarifiy

[Response] Thank you for your notification. I agree with it, and amended the sentence to make sense.

  1. add few line in introduction about explaining review article details

[Response] we added some description about it. [line 65-67]

  1. section 5 and 6 add few related article how role of antioxidant in human body, author should explain mechanism of oxidative stress in human voice with citiation and role of antioxidant agent on human body particularly vocal cord

[Response] The suggestion is very important, but in fact the mechanism of how antioxidant works specifically on vocal fold remains unclear with lack of data. We added just one citation (#85) to possibly explain this aspect [line 274-276]. This aspect should be the future task.

Reviewer 2 Report

Comments and Suggestions for Authors

Preclinical experimentation with the use of laboratory animals for scientific purposes is currently the most suitable method for obtaining meaningful and explanatory results for subsequent translation onto the human model. To date, oxidative stress is influential in many causes that affect some organs and/or tissues. In this regard, the well-executed and well-described manuscript highlights the aggravating role of bone stress on lesions of the vocal cords, and indicates which remedy point can be improved for the non-subsequent worsening of the lesion, through antioxidants which to date are been well studied as seen in the international literature.

Author Response

Thank you very much for taking the time to review this manuscript.

I really appreciate the reviewer’s thoughtful comments and insights about scientific research and translation to clinical commitment. We really agree that preclinical experiments using laboratory and animals are essential to develop new treatments for human patients, and keep going on the way.

Reviewer 3 Report

Comments and Suggestions for Authors

The authors of the manuscript present a brief report on the impact of oxidative stress on the voice and its maintenance with the help of antioxidants.

The manuscript is well documented (91 bibliographic references are cited) and structured in clear and concise, easily approaching and reading chapters.

line 24: it

My suggestion would be to detail the results of using the Twendee X product, describing in more detail the components of the product and the mechanisms by which they act.

I have no other comments to make, I congratulate the authors for developing this manuscript and bringing it to the attention of interested readers.

Author Response

[Response] Thank you for the very important suggestion. In fact, the mechanism of how Twendee X works for each organ has not fully been elucidated. We described some mechanism of how Twendee works for prevention of dementia in line 323-334, but there is no data or citation that can explain the specific mechanism of Twendee for vocal fold.

We added such description in line 334-336.